# The familial risk of infection-related hospitalization in children: A population-based sibling study

**Jessica E. Miller** [1,2]*, **Kim W. Carter**[3], **Nicholas de Klerk**[3], **David P. Burgner**[1,2,4]

**1** Murdoch Children's Research Institute, Royal Children's Hospital, Parkville, Victoria, Australia, **2** Department of Paediatrics, University of Melbourne, Parkville, Victoria, Australia, **3** Telethon Kids Institute, University of Western Australia, Perth, Western Australia, Australia, **4** Department of Paediatrics, Monash University, Clayton, Victoria, Australia

* jessica.miller@mcri.edu.au

**Data Availability Statement:** Data are available from the Western Australia Department of Health Data Linkage Branch with ethical approval through the Western Australia Department of Health Human Research Ethics Committee. To maintain

## Abstract

### Objective

To assess the risk of severe childhood infections within families, we conducted a sibling analysis in a population-based cohort study with genealogical linkage. We investigated the sibling risk of hospitalization with common infections, a marker of severity. We hypothesized that having siblings hospitalized for infection would increase the proband's risk of admission with infection.

### Study design

We used population data on Western Australian live-born singletons and their siblings between 1980 and 2014. Measures of infection were infection-related hospitalizations from discharge diagnostic codes. Exposure was having a sibling who had an infection-related hospitalization. Outcomes were infection-related hospitalizations in the child/proband. Probands were followed until an infection-related hospitalization admission (up to the first three), death, 18th birthday, or end of 2014, whichever occurred first. Infection risks were estimated by adjusted Cox proportional hazard models for multiple events.

### Results

Of 512,279 probands, 142,915 (27.9%) had infection-related hospitalizations; 133,322 (26.0%) had a sibling with a previous infection-related hospitalization (i.e. exposed). Median interval between sibling and proband infection-related hospitalizations was 1.4 years (interquartile range 0.5–3.7). Probands had a dose-dependent increase in risk if sibling/s had 1, 2, or 3+ infection-related hospitalizations (adjusted hazard ratio, aHR 1.41, 95% CI 1.39–1.43; aHR 1.65, 1.61–1.69; aHR 1.83, 1.77–1.90, respectively). Among siblings with the same clinical infection type, highest sibling risks were for genitourinary (aHR 2.06, 1.68–2.53), gastrointestinal (aHR 2.07, 1.94–2.19), and skin/soft tissue infections (aHR 2.34, 2.15–2.54). Overall risk of infection-related hospitalization was higher in children with more siblings and with older siblings.

confidentiality and security, interested individuals would need to apply for access to linked data by contacting the Western Australian Data Linkage Branch https://www.datalinkage-wa.org.au/.

**Funding:** This work is supported by the Australian National Health and Medical Research Council GTN572742, GTN106594, and GTN1064829 and GTN1175744. Research at Murdoch Children's Research Institute is supported by the Victorian Government's Operational Infrastructure Support Program. The funders had no role in study design, data collection and analysis, decision to publish, or preparation of the manuscript.

**Competing interests:** The authors have declared that no competing interests exist.

**Abbreviations:** CI, Confidence interval; HR, Hazard ratio; IQR, Interquartile range; SES, Socio-economic status; WA, Western Australia.

## Conclusion

In this population-based study, we observed an increased risk of infection-related hospitalization in children whose siblings were previously hospitalized for infection. Public health interventions may be particularly relevant in families of children hospitalized with infection.

## Introduction

Infection is a predominant cause of childhood morbidity, mortality and health service utilization. US national estimates from 2000–2012 indicate that infections in children aged <19 years accounted for 24.5% of all hospitalizations [1]. In 2003, 43% of US children aged <1 year admitted to hospital had an infection-related diagnosis, incurring a total cost of $690 million [2]. No analogous current data are available from Australia.

Microbial factors, such as outbreaks and heightened pathogen virulence, partly explain the marked variation in infection severity following exposure to similar pathogens [3]. Sibling analyses can account for shared heritable and environmental factors and measure associations from exposures at different ages in siblings [4]. Sibling analyses provide robust estimates of shared heritable determinants of infection that do not reflect seasonal variation in pathogen epidemiology and virulence. In genetic epidemiology, sibling risk ratio, or relative recurrence risk, is the ratio of risk in a sibling to a general population risk. Few studies have estimated the sibling risk ratio of infectious diseases [5].

We utilized total population data from Western Australia (WA) to estimate the risk of hospitalizations for childhood infections within families. We hypothesized that having siblings hospitalized for infection would increase the proband's risk of admission with infection. We explored whether susceptibility varied by specific types of infections, the age and number of siblings, and the interval between sibling and proband infection-related hospitalizations.

## Methods

### Study population

Since 2011, WA has had an average growth rate of 1.81% and currently, has a population of ~2.6 million people [6]. We conducted a population-based cohort study of live-born children in WA between January 1st 1980 and December 31st 2014, identified from the WA Birth Registry. Data from the WA Midwives' Notification System, Birth Register, Death Register, and Hospital Morbidity Database System were linked by probabilistic matching at the Data Linkage Branch of the WA Department of Health. Siblings were identified and their data linked with the Family Connections System, which creates links for pedigree relationships based on birth, death and marriage registrations [7]. Single child families and families with twins or higher multiples were excluded from the study; multiples may share virulent pathogens causing more severe contemporaneous infection and genetically identical multiples might inflate heritability estimates derived from siblings alone [8]. Half-siblings were not included as it was unknown when they became siblings or whether they resided in the same household. Individuals of Aboriginal or Torres Strait Islander descent were not included, as causal pathways are likely to be qualitatively different and include substantial economic disadvantage [9].

The study was approved by the Department of Health Human Research Ethics Committee (HREC), Murdoch Children's Research HREC and University of WA HREC.

## Infection-related hospitalization (exposure and outcome)

Data on infection-related hospitalizations were obtained from the Hospital Morbidity Database System, which has complete coverage of private and public WA hospital admissions [10]. Discharge diagnostic codes are based on the International Classification of Diseases (ICD) versions 9 and 10-AM and have been repeatedly validated [10].

Infection-related hospitalization is a robust and widely accepted measure of infection severity and reflects hospital admissions due to infections of sufficient severity to warrant inpatient care [9, 11]. Infection-related hospitalizations were identified using principal and up to twenty additional ICD codes, a more sensitive measure of total infection burden than the principal code alone that has been used by other researchers for similar purposes [9, 12]. An infection-related hospitalization was defined as admission with ≥1 infection discharge code occurring ≥24 hours after discharge for the birth-related hospitalization. We restricted infection-related hospitalizations to readmission following the birth-related discharge to avoid bias from suspected neonatal sepsis that is often either unconfirmed or caused by direct microbial exposure during birth. Overlapping or nested hospital transfers, where <24 hours had elapsed between discharge and re-hospitalization, and repeat infection-related hospitalizations within 7 days were considered as single admissions.

ICD infection codes were categorized *a priori* into seven broad clinical groups: invasive bacterial, gastrointestinal, lower respiratory tract, upper respiratory tract, skin and soft tissue, genitourinary, and viral infections, as previously described [13]. Additionally, we assessed specific common infections: tonsillitis (acute and chronic), otitis media (chronic), gastroenteritis and bronchiolitis; acute otitis media was uncommon and not included.

The study outcome was infection-related hospitalization occurring in the child of analysis, termed 'proband' hereafter ('proband infection-related hospitalization'). Up to the first 3 proband infection-related hospitalizations occurring during the follow-up period were considered and analyzed as recurrent events data.

Exposure was defined as the number of infection-related hospitalizations occurring in a proband's sibling ('sibling infection-related hospitalization'), categorized as (0, 1, 2, 3+). The proband became 'exposed' once a sibling had an infection-related hospitalization during the follow-up period. Exposure was considered time-varying; once a proband became exposed, they were exposed for the remainder of the follow-up period and the level of exposure at the time of each outcome, was the number of sibling infection-related hospitalizations that occurred prior to that outcome. Probands were 'unexposed' for the period preceding the first sibling infection-related hospitalization or if no sibling infection-related hospitalizations occurred during follow-up.

## Covariates

Data on maternal age, smoking during pregnancy, parity, birth weight, gestational age, sex, Apgar score, and birth mode were obtained from the Midwives' Notification System, which collates details on all births in WA from 1980 onwards. Area-level socioeconomic status (SES) was derived from Socio-Economic Indexes for Areas (SEIFA), which are summary measures of socioeconomic variables associated with disadvantage at the census Collection District level. The indexes can be used to rank collection districts according to the general socioeconomic wellbeing of residents. Percentiles for SES were defined by matching address at birth to the SEIFA score for the same census Collector's District from the census year closest to the birth year [14]. Data on smoking during pregnancy were available from 1997 onwards.

## Statistical methods

We estimated sibling risk ratios as the risk of infection-related hospitalizations in exposed probands (i.e. those whose sibling/s had an infection-related hospitalization) compared to the risk in unexposed probands (i.e. those whose sibling/s had not had infection-related hospitalization/s). For all children, follow-up time began when they became a sibling: for children with older siblings, follow-up time began at the child's birth-related hospital discharge date; for children with only younger siblings, follow-up time began at the date of birth of their first sibling. All children were followed until they had an infection-related hospitalization admission (up to the first three admissions), death, 18th birthday, or end of 2014, whichever occurred first. Multivariable analyses adjusted for maternal age (<20, 20–24, 25–29, 30–34, ≥35 years), parity (previous births no/yes), gestational age (<28, 28–29, 30–31, 32–34, 35, 36, 37, 38, 39–40, 41, ≥42 weeks), birth weight (1000–1499, 1500–1999, 2000–2499, 2500–2999, 3000–3499, 3500–3999, ≥4000 grams), sex, season of birth (spring, summer, autumn, winter), 5-minute Apgar score (0–7, 8–10), birth mode (vaginal/cesarean), and year (2 year blocks). Adjusted hazard ratios (HRs) and 95% confidence intervals (CIs) were estimated using multivariate Cox proportional hazard regression models for recurrent events data (conditional risk set models) by Prentice et al. [15]. Proband infection-related hospitalizations were the events and age in months was the underlying time variable. Time to each event, up to the first 3 admissions, was measured from entry time, and analyses were stratified by event order with the final estimate as the overall effect. Children in a family generally contributed to the analyses both as a proband and as a sibling, when their sibling was the proband. Depending on family size, children could occur in more than one proband/sibling unit. Robust standard errors accounted for dependence of family members.

Data on smoking during pregnancy (no/yes) were not included in overall analyses, as these data were unavailable for the entire study period. Given the missing values for smoking and SES, sensitivity analyses were conducted including and excluding these variables. All other covariates had few missing data; no imputations were warranted.

We estimated sibling risk ratios for overall, clinical group, and specific common infections. For clinical and specific infection sibling risk ratios, risks were estimated among probands and siblings who presented with the same clinical group or specific infection. We additionally estimated clinical group estimates if siblings presented with any infection. For specific infections, we conducted stratified analyses for siblings younger or older than the proband. For simplicity and to explore an age effect, we restricted these analyses to families where siblings were either all younger or all older than the proband.

We calculated overall infection-related hospitalization risk in children age <5 and 5–18 years by the time-at-risk for infection-related hospitalization following exposure. Time-at-risk was defined among the unexposed as the time from the start of follow-up to the first proband infection-related hospitalization or end of study, whichever occurred first, and among the exposed as the time from the first sibling infection-related hospitalization to the first proband infection-related hospitalization or end of study, whichever occurred first. Within time-at-risk intervals and exposure group, infection-related hospitalization risk was calculated as number of cases divided by total number unexposed or exposed, respectively.

We estimated sibling risk ratios by family size (1, 2, 3+ siblings) and by age of siblings (younger/older than proband), accounting for the number and distribution of the sibling infection-related hospitalizations. Analyses by age of sibling were restricted to families where all siblings were either younger or older than the proband. To assess the possible contribution of pathogen sharing, we calculated the time between admission in the sibling and first admission in the proband with the same clinical infection type. To minimize risk due to pathogen

sharing among siblings, we also performed analyses excluding those whose proband infection-related hospitalization admissions occurred within 30 days of the sibling exposure.

We performed a sensitivity analysis using infection-related hospitalizations identified by the principal and first 3 additional ICD codes. To address potential unmeasured confounding, we calculated an E-value, which estimates the minimum strength of association that an unmeasured confounder would need to have with both the exposure and outcome in order to fully explain away the observed association [16]. Lastly, we calculated population attributable fractions for all infections and for bronchiolitis. Analyses were performed in STATA 13.0 [17].

## Results

Of the 512,279 probands in the study population, 369,364 (72.1%) did not have an infection-related hospitalization and 142,915 (27.9%) had at least one infection-related hospitalization, of which 49,582 (9.7% of the study population) had multiple infection-related hospitalization. By the end of follow-up, 133,322 of the study population (26.0%) were exposed, i.e. had a sibling with an infection-related hospitalization (Fig 1). On average, exposed probands were first-born children with younger mothers (Table 1). The median age of when probands were

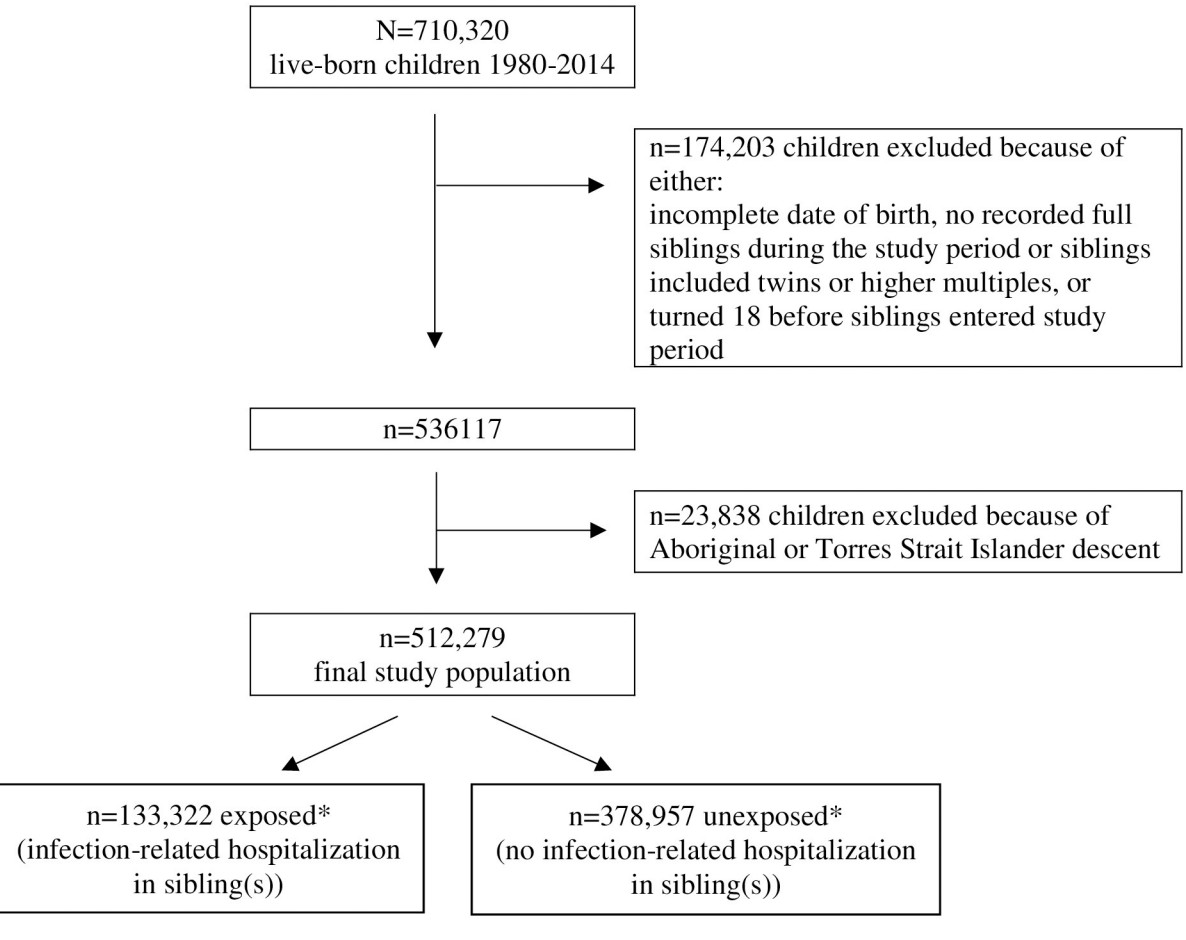

*Time-varying exposure status at end of follow-up period

**Fig 1. Flowchart of study population.**

**Table 1. Characteristics of proband children in study population, time-varying exposure status at end of follow-up period.**

| Characteristic | Infection-related hospitalization in sibling(s) (Exposure) | | | | Total study population | | Chi-squared p-value[*] |
| --- | --- | --- | --- | --- | --- | --- | --- |
| | No (Unexposed) | | Yes (Exposed) | | | | |
| | N | % | N | % | N | % | |
| | **378,957** | **74** | **133,322** | **26** | **512,279** | **100** | |
| **Maternal age at birth** | | | | | | | |
| <20 | 10,619 | 2.8 | 6,151 | 4.6 | 16770 | 3.27 | <0.0001 |
| 20–24 | 66,727 | 17.6 | 31,177 | 23.4 | 97904 | 19.11 | |
| 25–29 | 131,256 | 34.6 | 49,307 | 37.0 | 180563 | 35.25 | |
| 30–34 | 117,912 | 31.1 | 34,798 | 26.1 | 152710 | 29.81 | |
| ≥35 | 52,440 | 13.8 | 11,888 | 8.9 | 64328 | 12.56 | |
| Missing | 3 | 0.0 | 1 | 0.0 | 4 | 0.0 | |
| **Maternal parity (had previous birth(s))** | | | | | | | |
| No | 141,861 | 37.4 | 61,083 | 45.8 | 202,944 | 39.6 | <0.0001 |
| Yes | 237,096 | 62.6 | 72,239 | 54.2 | 309,335 | 60.4 | |
| Missing | 0 | 0.0 | 0 | 0.0 | 0 | 0.0 | |
| **Sex** | | | | | | | |
| Male | 197,026 | 52.0 | 66,469 | 49.9 | 263,495 | 51.4 | <0.0001 |
| Female | 181,927 | 48.0 | 66,853 | 50.1 | 248,780 | 48.6 | |
| Missing | 4 | 0.0 | 0 | 0.0 | 4 | 0.0 | |
| **Season of birth** | | | | | | | |
| Spring | 94,813 | 25.0 | 33,223 | 24.9 | 128,036 | 25.0 | 0.707 |
| Summer | 91,339 | 24.1 | 32,245 | 24.2 | 123,584 | 24.1 | |
| Autumn | 97,757 | 25.8 | 34,276 | 25.7 | 132,033 | 25.8 | |
| Winter | 95,048 | 25.1 | 33,578 | 25.2 | 128,626 | 25.1 | |
| Missing | 0 | 0.0 | 0 | 0.0 | 0 | 0.0 | |
| **Apgar score 5 minutes** | | | | | | | |
| 0–7 | 9,899 | 2.6 | 3,597 | 2.7 | 13,496 | 2.6 | 0.146 |
| 8–10 | 368,796 | 97.3 | 129,644 | 97.2 | 498,440 | 97.3 | |
| Missing | 262 | 0.1 | 81 | 0.1 | 343 | 0.0 | |
| **Birth mode** | | | | | | | |
| Vaginal | 283,643 | 74.9 | 102,430 | 76.8 | 386,073 | 75.4 | <0.0001 |
| Cesarean | 95,314 | 25.2 | 30,892 | 23.2 | 126,206 | 24.6 | |
| Missing | 0 | 0.0 | 0 | 0.0 | 0 | 0.0 | |
| **Birth year** | | | | | | | |
| 1980–84 | 30,956 | 8.2 | 13,528 | 10.2 | 44,484 | 8.7 | <0.0001 |
| 1985–89 | 51,432 | 13.6 | 22,209 | 16.7 | 73,641 | 14.4 | |
| 1990–94 | 56,531 | 14.9 | 24,588 | 18.4 | 81,119 | 15.8 | |
| 1995–99 | 57,653 | 15.2 | 24,155 | 18.1 | 81,808 | 16.0 | |
| 2000–04 | 58,840 | 15.5 | 21,452 | 16.1 | 80,292 | 15.7 | |
| 2005–10 | 87,209 | 23.0 | 22,934 | 17.2 | 110,143 | 21.5 | |
| 2011–13 | 36,336 | 9.6 | 4,456 | 3.3 | 40,792 | 8.0 | |
| Missing | 0 | 0.0 | 0 | 0.0 | 0 | 0.0 | |

(*Continued*)

**Table 1.** (Continued)

| Characteristic | Infection-related hospitalization in sibling(s) (Exposure) | | | | Total study population | | Chi-squared p-value* |
|---|---|---|---|---|---|---|---|
| | No (Unexposed) | | Yes (Exposed) | | | | |
| | N | % | N | % | N | % | |
| | 378,957 | 74 | 133,322 | 26 | 512,279 | 100 | |
| **Gestational age (weeks)** | | | | | | | |
| <28 | 533 | 0.1 | 135 | 0.1 | 668 | 0.1 | <0.0001 |
| 28–29 | 592 | 0.2 | 187 | 0.1 | 779 | 0.2 | |
| 30–31 | 969 | 0.3 | 319 | 0.2 | 1,288 | 0.3 | |
| 32–34 | 4,635 | 1.2 | 1,615 | 1.2 | 6,250 | 1.2 | |
| 35 | 4,461 | 1.2 | 1,703 | 1.3 | 6,164 | 1.2 | |
| 36 | 9,641 | 2.5 | 3,473 | 2.6 | 13,114 | 2.6 | |
| 37 | 28,210 | 7.4 | 9,892 | 7.4 | 38,102 | 7.4 | |
| 38 | 81,052 | 21.4 | 26,706 | 20.0 | 107,758 | 21.0 | |
| 39–40 | 195,890 | 51.7 | 69,037 | 51.8 | 264,927 | 51.7 | |
| 41 | 47,410 | 12.5 | 17,814 | 13.4 | 65,224 | 12.7 | |
| ≥42 | 5,145 | 1.4 | 2,232 | 1.7 | 7,377 | 1.4 | |
| Missing | 419 | 0.1 | 209 | 0.2 | 628 | 0.1 | |
| **Birthweight (grams)** | | | | | | | |
| <1000 | 564 | 0.2 | 141 | 0.1 | 705 | 0.1 | <0.0001 |
| 1000-<1500 | 1,120 | 0.3 | 384 | 0.3 | 1,504 | 0.3 | |
| 1500-<2000 | 2,342 | 0.6 | 814 | 0.6 | 3,156 | 0.6 | |
| 2000-<2500 | 9,605 | 2.5 | 3,575 | 2.7 | 13,180 | 2.6 | |
| 2500-<3000 | 53,240 | 14.1 | 19,384 | 14.5 | 72,624 | 14.2 | |
| 3000-<3500 | 144,371 | 38.1 | 51,146 | 38.4 | 195,517 | 38.2 | |
| 3500-<4000 | 123,930 | 32.7 | 43,106 | 32.3 | 167,036 | 32.6 | |
| ≥4000 | 43,779 | 11.6 | 14,769 | 11.1 | 58,548 | 11.4 | |
| Missing | 6 | 0.0 | 3 | 0.0 | 9 | 0.0 | |
| **Socioeconomic status (percentile)** | | | | | | | |
| lowest <10 | 23,606 | 6.2 | 10,071 | 7.6 | 33,677 | 6.6 | <0.0001 |
| 10–25 | 46,252 | 12.2 | 17,538 | 13.2 | 63,790 | 12.5 | |
| 25–50 | 83,934 | 22.2 | 30,553 | 22.9 | 114,487 | 22.4 | |
| 50–75 | 88,923 | 23.5 | 29,572 | 22.2 | 118,495 | 23.1 | |
| 75–90 | 61,186 | 16.2 | 19,125 | 14.3 | 80,311 | 15.7 | |
| highest >90 | 42,552 | 11.2 | 12,881 | 9.7 | 55,433 | 10.8 | |
| Missing | 32,504 | 8.6 | 13,582 | 10.2 | 46,086 | 9.0 | |
| **Maternal smoking during pregnancy** | | | | | | | |
| No | 184,327 | 48.6 | 51,065 | 38.3 | 235,392 | 46.0 | <0.0001 |
| Yes | 24,816 | 6.6 | 8,610 | 6.5 | 33,426 | 6.5 | |
| Missing | 169,814 | 44.8 | 73,647 | 55.2 | 243,461 | 47.5 | |
| **Total number of infection-related hospitalizations during study period in proband** | | | | | | | |
| 0 | 263,982 | 69.7 | 105,382 | 79.0 | 369,364 | 72.1 | <0.0001 |
| 1 | 73,930 | 19.5 | 19,403 | 14.6 | 93,333 | 18.2 | |
| 2 | 23,469 | 6.2 | 5,273 | 4.0 | 28,742 | 5.6 | |
| ≥3 | 17,576 | 4.6 | 3,264 | 2.5 | 20,840 | 4.1 | |

*Chi-square test of independence for exposure status and dichotomous and categorized measures.

exposed was 4.7 years (interquartile range, IQR 2.7–7.7) among those who did not have an infection-related hospitalization during follow-up, and 4.2 (IQR 2.2–7.1) among those who did. The median age in probands for their first infection-related hospitalization was 3.2 years (IQR 1.2–6.0); among exposed 5.5 years (IQR 3.0–9.1) and among unexposed 2.8 (IQR 1.1–5.4). The median time between a sibling and proband infection-related hospitalization was 1.4 years (IQR 0.5–3.7).

We observed an increased risk of infection-related hospitalization among probands whose sibling/s had infection-related hospitalization/s. The risk increased in a dose-dependent manner with number of infection-related hospitalizations in the sibling/s (1, 2, 3+) (adjusted hazard ratio, aHR 1.41, 95% CI 1.39–1.43; aHR 1.65, 1.61–1.69; aHR 1.83, 1.77–1.90, respectively) (Fig 2A). Risk persisted even when proband infection-related hospitalization occurring within 30 days of a sibling infection-related hospitalization were excluded (Fig 2B).

Among families where the proband and their sibling were hospitalized with the same clinical infection type, the median time between sibling and proband hospitalizations ranged from 0.7 years (IQR 0.01–2.9) for gastrointestinal infections, to 2.9 (IQR 1.5–5.9) for invasive bacterial infections. The highest risks were for genitourinary (aHR 2.06, 1.68–2.53), gastrointestinal (aHR 2.07, 1.94–2.19) and skin and soft tissue infections (aHR 2.34, 2.15–2.54). Risks were lower for any infection than for the same clinical infection group (Fig 3).

For the specific infections, the median time between sibling and proband infection-related hospitalization ranged from 40 days for bronchiolitis to 2.4 years for chronic tonsillitis. The highest observed risk was for chronic tonsillitis among probands who had an older sibling previously hospitalized for chronic tonsillitis (aHR 3.08, 2.92–3.24) (S1 Fig).

Overall risks of infection were highest among preschool children (age <5 years) (Fig 4). For all children, the highest risk for an infection-related hospitalization occurred among children whose sibling was hospitalized with an infection within the previous month (0.93 in exposed children <5 years, 0.63 in exposed children 5–18 years). In children <5 years, the risk difference between exposed and unexposed children remained fairly constant regardless of when, in the 12 months prior to a proband infection-related hospitalization, the exposure occurred. In children aged 5–18 years, the risk difference between exposed and unexposed was similar regardless of whether exposure occurred one month to 2 years prior to a proband infection-related hospitalization.

The dose-response effect with increasing number of sibling infection-related hospitalizations persisted despite family size. However, with increasing family size, overall infection-related hospitalization risks slightly decreased. Overall risk of infection was slightly higher in children with older siblings (Fig 5A). Increased infection-related hospitalization risk among exposed probands persisted even when proband infection-related hospitalization occurring within 30 days of a sibling infection-related hospitalization were excluded (Fig 5B). In families with 2+ siblings, we did not observe any pattern of association with sibling infections occurring in one or multiple siblings (Fig 5C).

Sensitivity analyses including only infections coded as the principal or first three additional diagnostic codes showed similar results (S1 Table). Based on the E-value, an unmeasured confounder would need an association of ≥1.85 with infection-related hospitalization in both the sibling and proband, to explain away the observed associations, but weaker confounding could not.

We calculated that 12.2% (11.8–12.6) (n = 29027) of infection-related hospitalizations in the population would be potentially prevented if siblings did not have prior infection-related hospitalizations. If multiple infection-related hospitalizations in siblings were reduced, 4.4% (4.2–4.6) (n = 10,469) of infection-related hospitalizations in probands could be prevented.

**A)**

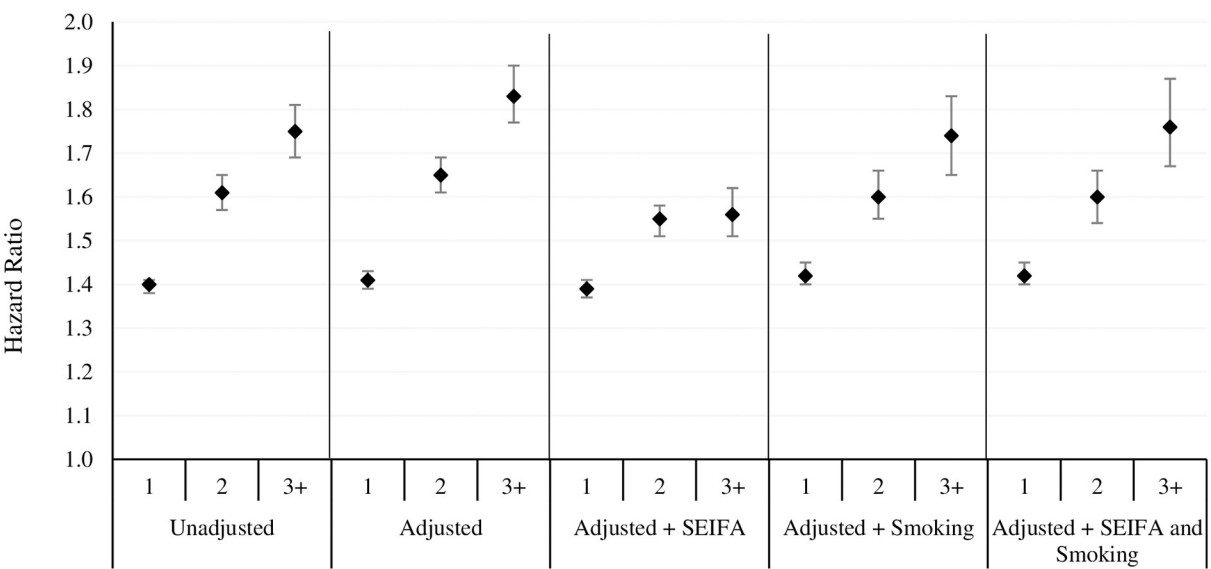

**B)**

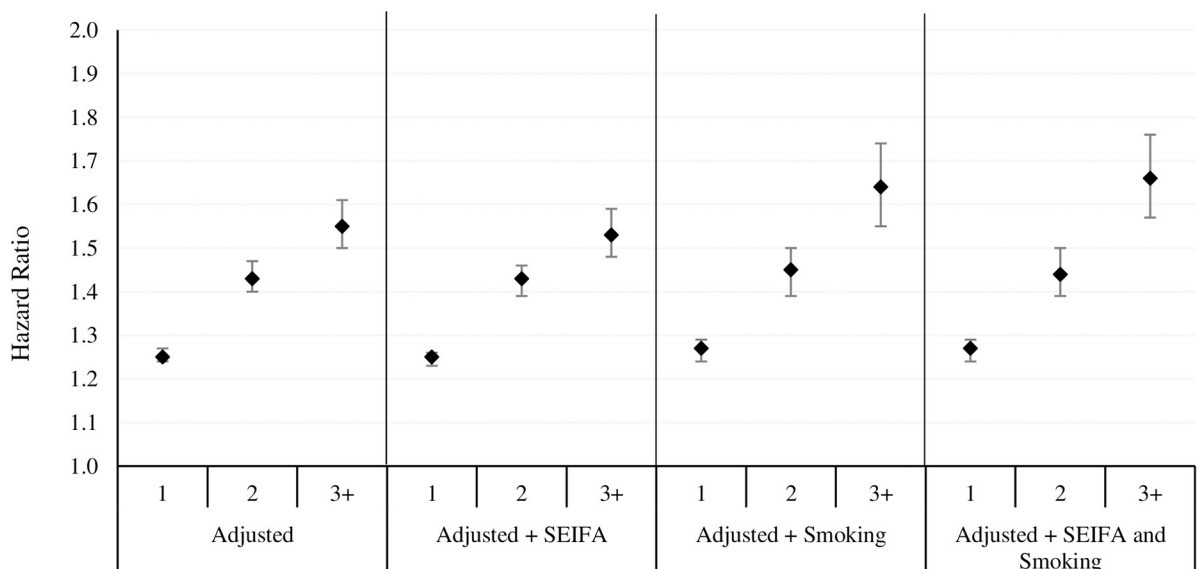

**Fig 2. Overall infection-related hospitalization sibling hazard ratios.** Adjusted for maternal age, parity, gestational age, birth weight, sex, season of birth, 5-minute Apgar score, birth mode, and birth year. Additional adjustments for SES (from Socio-Economic Index for Areas (SEIFA) data) and smoking during pregnancy. Smoking data available from 1997–2013. (A) Infection-related hospitalization sibling hazard ratios among probands whose sibling/s had infection-related hospitalizations compared to probands whose siblings/s did not have infection-related hospitalizations (reference group). (B) Infection-related hospitalization sibling hazard ratios excluding proband infection-related hospitalization admissions occurring within 30 days following a sibling infection-related hospitalization exposure.

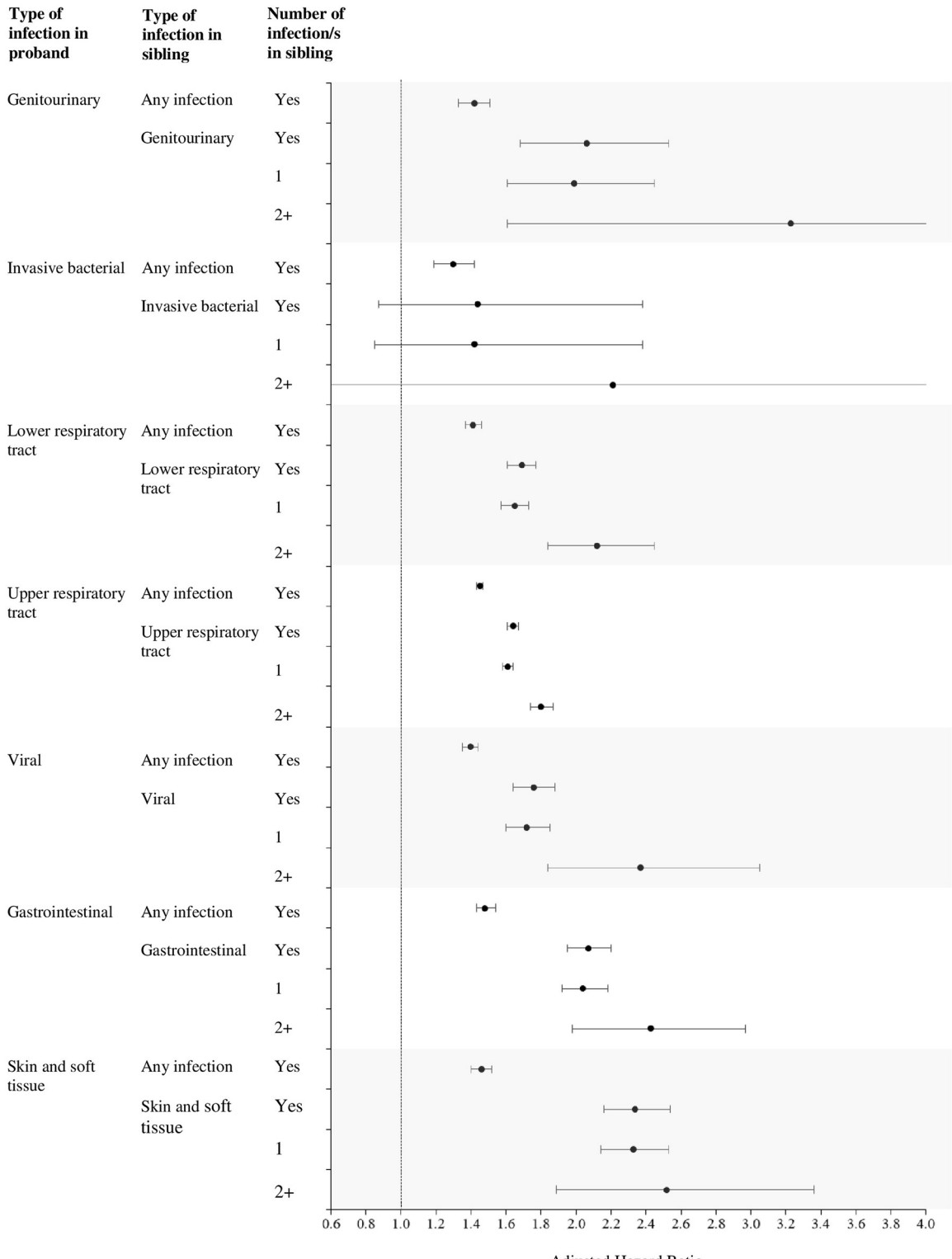

**Fig 3. Infection-related hospitalization sibling hazard ratios by clinical infection groups.** Infection-related hospitalization sibling hazard ratios by type of clinical infection in proband (outcome) and sibling (exposure), compared to risk in probands whose sibling/s did not have infection-related hospitalizations (reference group). Adjusted for maternal age, parity, gestational age, birth weight, sex of the child, season of birth, 5-minute Apgar score, birth mode, birth year. Number of infections in sibling was analyzed as both dichotomous (Yes/No (reference)) and categorical (0 (reference), 1, 2+).

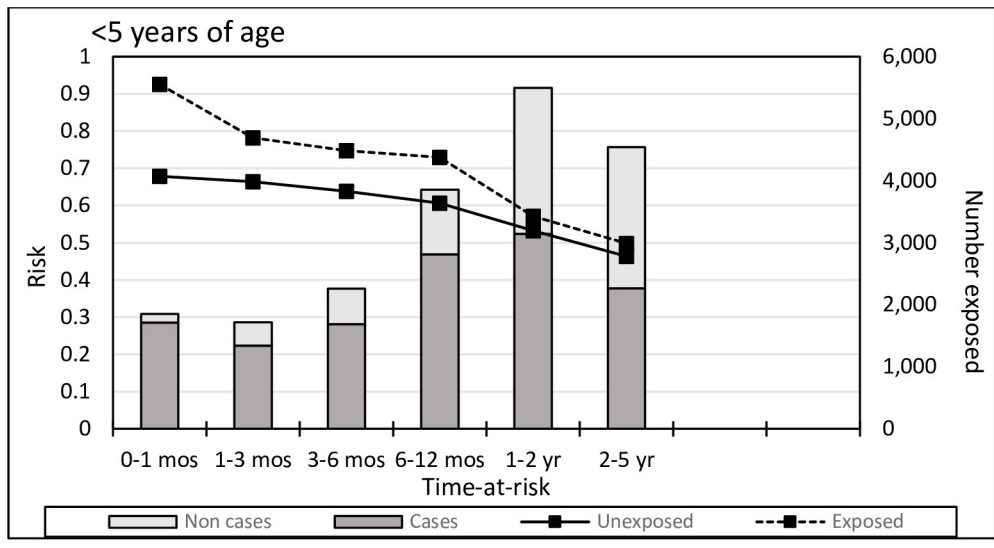

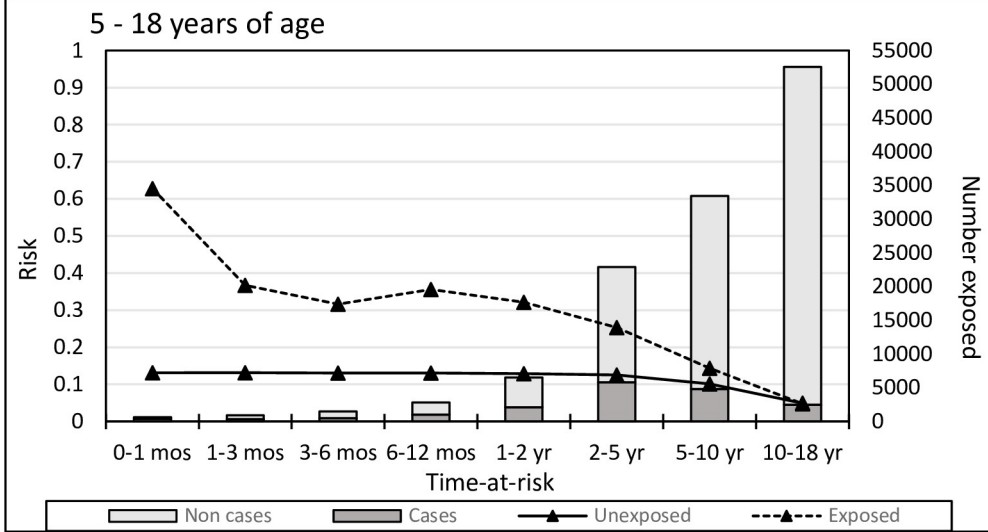

**Fig 4. Risk of 1ˢᵗ occurrence infection-related hospitalization in children <5 and 5–18 years old, by time-at-risk.**
In the exposed group, 'time-at-risk' is the time from the first sibling infection-related hospitalization (occurring prior
to the 1ˢᵗ proband infection-related hospitalization) to the 1ˢᵗ proband infection-related hospitalization or end of study,
whichever occurred first. In the unexposed group, 'time-at-risk' begins with the start of follow-up. Follow-up times
shorter than the 'time-at-risk' interval were not included in the interval risk calculations. 'Cases' refers to probands
with infection-related hospitalizations; 'Non cases' refers to probands with no infection-related hospitalizations.

1.3% (0.5–2.1) (n = 200) of infection-related hospitalizations for bronchiolitis would be pre-
vented if siblings did not have bronchiolitis.

## Discussion

The risk of hospitalization with infection was increased among children whose siblings had
previous infection-related hospitalizations, compared with children whose siblings had no pre-
vious infection-related hospitalizations. The risk increased in a dose-response manner from
40% in children with one sibling infection-related hospitalization to 80% with ≥3 sibling infec-
tion-related hospitalizations.

A)

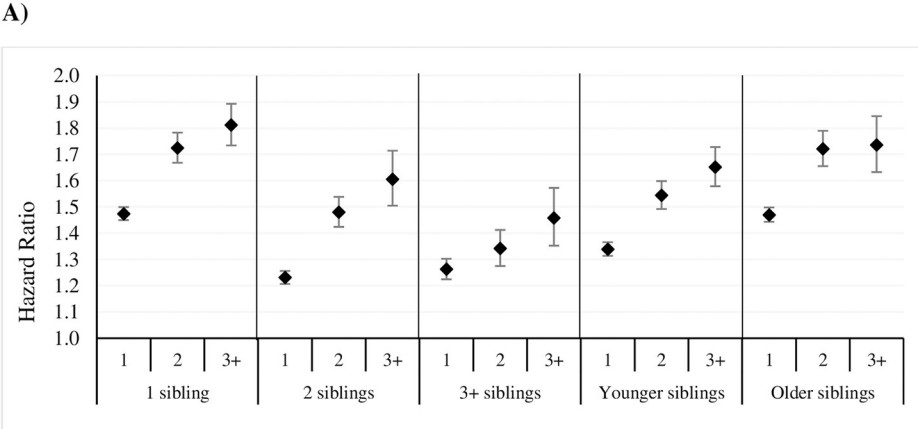

B)

C)

**Fig 5. Infection-related hospitalization sibling hazard ratios by family size and sibling characteristics.** Adjusted for maternal age, parity, gestational age, birth weight, sex of the child, season of birth, 5-minute Apgar score, birth mode, birth year. (A) Infection-related hospitalization sibling hazard ratios by family size and age of siblings. (B) Infection-related hospitalization sibling hazard ratios excluding proband infection-related hospitalization admissions occurring within 30 days following a sibling infection-related hospitalization exposure. (C) Infection-related

hospitalization sibling hazard ratio by family size and distribution of infections among siblings. IRH = infection-related hospitalization.

Increased risks were observed for all clinical and specific infection groups. The long median interval (1.4 years) between sibling and proband infection-related hospitalization suggests that factors other than sharing of virulent pathogens or seasonal outbreaks explain the increased risks [8]. Further, when sibling and proband had infection-related hospitalizations in the same clinical group, risks increased, suggesting shared underlying risks, such as more specific heritable and environmental factors [3, 18].

Sibling infection-related hospitalization risk was reduced in larger families, but the mechanisms are unclear. This may reflect differential use of child care, a well-recognized risk factor for childhood infection [19]; families with more older children use less child care. Almost 50% of Australian families with ≤2 children are likely to be enrolled in child care compared to 33% of larger families (3+ children) [20], however, data on child care attendance in our population were unavailable. In addition, larger sibship size is protective against asthma, allergy and some autoimmune diseases in children and adults [21, 22]. Early microbial exposures associated with increased family size, such as increased microbiome diversity, may optimise early life immune responses [23]. Overall risks were slightly lower in families with siblings younger than the proband, likely reflecting reduced susceptibility to more severe infection with increasing age.

The strengths of our study include 35 years of total population linked data with minimal loss to follow-up. The WA Data Linkage System is a powerful and unique resource of data from a stable population [24]. All WA hospital admission data are captured and almost all maternal full siblings have been identified. The Midwives' Notification Form records >99% of all live births [25]. Few other studies have examined sibling risks of infectious diseases apart from those reporting on single infectious diseases [5].

Our outcome measure of infection-related hospitalization is a marker of infectious disease severity. Hospitalization is less influenced by health-seeking behavior, social disadvantage and physician management than emergency department or primary care presentations and by practitioner-related variation in management [26, 27]; further studies using these less severe outcomes of infection are of interest, however these data were unavailable for this study. Inclusion of principal and additional diagnostic codes, an approach used in analogous studies [12], captures infection burden more completely than the principal code alone. Sensitivity analyses that captured infections based on the principal and first three additional codes showed similar results as when all 20 additional codes were considered.

Sibling studies are more powerful than twin studies for less common but clinically important infections [3]. Infections in sibling pairs (unlike twins) generally occur many months apart, much longer than the incubation period for almost all childhood pathogens. This reduces possible biases from over-diagnosis in the second child, from clinically unwarranted hospitalization with the same infection in the second child (e.g. social factors or parental anxiety), and from the confounding effects of shared virulent pathogens. Although many infections show increased concordance rates in monozygous versus dizygous twins [18, 28], twin infection studies may overestimate genetic factors and underestimate shared environmental exposures. For example, twin data on tuberculosis, where increased concordance in monozygous twins has long been suggestive of host genetic susceptibility, have been re-analyzed and the findings are suggested to be largely due to environmental factors which may outweigh the importance of hereditary factors [29].

Population analyses have some unavoidable limitations. Data are limited to routinely collected and recorded information. Data on infections managed in primary care or in emergency departments and data on potential unmeasured co-variates, (e.g. child care attendance, breast-feeding, tobacco smoke exposure, household structure, obesity, parental chronic disease, and environmental exposures) were unavailable. Smoking data were available from 1997 onwards; SES data were missing for 9% of the population. We therefore present our main findings with and without adjustments for smoking and SES. We could not assess migration from WA or transfer to facilities outside the state; both are rare [30]. Family Connections uses registrations where parents may not be biological (e.g. adoptions, surrogacy), but these families could not be identified. The estimated small percentage of adoptions is unlikely to skew our results but should be considered when interpreting heritability.

Sibling risks reflect shared genomic susceptibility and environmental exposures. However, the long interval between sibling and proband admission suggests infection within families with virulent organisms is an unlikely explanation for increased risks, especially as most pediatric pathogens are widely distributed and carried asymptomatically in the majority [3]. In a study of meningococcal disease, the sibling risk ratio was 30.3 overall and 8.2 if cases occurred >1 year apart [8], indicating that a more accurate estimate of the contribution of shared heritable and environmental factors, independent of pathogen sharing, is likely if the interval between cases is longer than incubation period and asymptomatic carriage of the invasive strain. In our study, we performed a sub-group analysis excluding cases within 30 days following the sibling's infection and the increased risk in the proband remained.

Changes in healthcare practice in Australia since the end of the study in 2014 are unlikely to have influenced study results. Apart from universal free influenza immunization for children aged 6 months to 5 years in 2018, there have been few changes to public health interventions that would affect the data. Other changes to immunization policy, such as introduction of meningococcal B vaccine are unlikely to have impacted the findings as the incidence is very low. Expansion of hospital in the home antibiotic therapy (outpatient antibiotic therapy in the US) has expanded in the last few years, but these children are still classified as hospital admissions and so the outcome data would not be affected.

The study, which may be broadly applicable to other high-income settings, highlights the increased risk in siblings of children hospitalized for infection. Interventions for other family members in this context is rarely considered in clinical pediatric practice (beyond specific infections, such as antibiotic prophylaxis for meningococcal infection). These findings suggest that simple interventions, such as promoting breast-feeding of younger siblings and timely and complete vaccination may be particularly pertinent in families where a child has been hospitalized with an infection.

## Conclusions

These data help establish the heritable association of hospitalization for common childhood infections among siblings. The findings highlight the importance of preventative measures and efforts to reduce environmental risk factors, especially in families of children hospitalized with infection.

## Supporting information

**S1 Fig. Infection-related hospitalization sibling hazard ratios for specific infection groups.** Infection-related hospitalization sibling hazard ratios among probands whose sibling/s had infection-related hospitalization/s for the same specific infection compared to risk in probands whose siblings/s did not have infection-related hospitalizations (reference group), among

younger and older siblings. Adjusted for maternal age, parity, gestational age, birth weight, sex, season of birth, 5-minute Apgar score, birth mode, and birth year. ICD 9 and 10 codes for bronchiolitis: 466.1, J21; gastroenteritis: 009, 535.4 535.7, A09; otitis media chronic: 381.1, 381.2, 381.3, 381.4, 382.1, 382.2, 382.3, H65.2, H65.3, H65.4, H65.9, H66.1, H66.2, H66.3; tonsillitis acute: 034.0, 463, J03; tonsillitis chronic: 474, J35. Acute otitis media 381.0, 382.0, H65.0, H65.1, H66.0 was uncommon and not included.
(PDF)

**S2 Fig. Time from exposure (sibling infection-related hospitalization) to outcome (proband infection-related hospitalization) for all and specific infection groups.**
(PDF)

**S1 Table. Sensitivity analysis of infection-related hospitalization identification.**
(PDF)

**S1 Checklist. STROBE statement—Checklist of items that should be included in reports of *cohort studies.***
(DOCX)

## Acknowledgments

The authors would like to thank the staff at the Western Australia Data Linkage Branch. We thank the data custodians of the Birth Registrations, Death Registrations, Hospital Morbidity Data Collection and Midwives Notification System.

## Author Contributions

**Conceptualization:** Kim W. Carter, Nicholas de Klerk, David P. Burgner.

**Formal analysis:** Jessica E. Miller, Nicholas de Klerk.

**Funding acquisition:** Kim W. Carter, Nicholas de Klerk, David P. Burgner.

**Methodology:** Jessica E. Miller, Kim W. Carter, Nicholas de Klerk, David P. Burgner.

**Project administration:** Jessica E. Miller.

**Software:** Kim W. Carter.

**Supervision:** Kim W. Carter, Nicholas de Klerk, David P. Burgner.

**Validation:** Jessica E. Miller.

**Visualization:** Jessica E. Miller, Nicholas de Klerk, David P. Burgner.

**Writing – original draft:** Jessica E. Miller, David P. Burgner.

**Writing – review & editing:** Jessica E. Miller, Kim W. Carter, Nicholas de Klerk, David P. Burgner.

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
