## [Decision Letter · Decision Letter 0]

1 Dec 2020

PONE-D-20-33021

The familial risk of infection-related hospitalization: a population-based sibling study

PLOS ONE

Dear Dr. Miller,

Thank you for submitting your manuscript to PLOS ONE. After careful consideration, we feel that it has merit but does not fully meet PLOS ONE’s publication criteria as it currently stands. Therefore, we invite you to submit a revised version of the manuscript that addresses the points raised during the review process.

We look forward to receiving your revised manuscript.

Kind regards,

Sreeram V. Ramagopalan

Academic Editor

PLOS ONE

Journal Requirements:

2. We note that you have reported significance probabilities of 0 in places. Since p=0 is not strictly possible, please correct this to a more appropriate limit, eg 'p<0.0001'.

4. Please ensure that you refer to Figure 1 in your text as, if accepted, production will need this reference to link the reader to the figure.

Reviewers' comments:

Reviewer's Responses to Questions

**Comments to the Author**

1. Is the manuscript technically sound, and do the data support the conclusions?

Reviewer #1: Yes

2. Has the statistical analysis been performed appropriately and rigorously? 

Reviewer #1: Yes

3. Have the authors made all data underlying the findings in their manuscript fully available?

Reviewer #1: Yes

4. Is the manuscript presented in an intelligible fashion and written in standard English?

Reviewer #1: Yes

5. Review Comments to the Author

Reviewer #1: This is a well-written manuscript for a well-conducted study. The manuscript may benefit from revisions as indicated in the comment bubbles inserted in the attached files to improve the clarity and impact on intended audience.

6. PLOS authors have the option to publish the peer review history of their article (what does this mean?). If published, this will include your full peer review and any attached files.

Reviewer #1: No

---

## [Author Response · Author response to Decision Letter 0]

12 Jan 2021

Journal Requirements:

RESPONSE: We have made sure that the manuscript meets the style requirements. Specifically, we have placed all figure captions directly after the paragraph in which they are first mentioned. 

2. We note that you have reported significance probabilities of 0 in places. Since p=0 is not strictly possible, please correct this to a more appropriate limit, eg 'p<0.0001'.

RESPONSE: We have made the suggested change.

RESPONSE: Data for this study are not available due to legal and ethical restrictions imposed by the Department of Health of Western Australia and the Western Australian Human Research Ethics Committee. The data are considered privileged and confidential and all authors signed declarations stating they would not provide any of the data to any other person or institution. Therefore unauthorized persons do not have access to the data either physically or electronically. We have updated the Data Availability statement to:

Data are available from the Western Australia Department of Health Data Linkage Branch with ethical approval through the Western Australia Department of Health Human Research Ethics Committee. To maintain confidentiality and security, interested individuals would need to apply for access to linked data by contacting the Western Australian Data Linkage Branch https://www.datalinkage-wa.org.au/

4. Please ensure that you refer to Figure 1 in your text as, if accepted, production will need this reference to link the reader to the figure.

RESPONSE: We have now referenced Figure 1 in the results section:

PAGE 9, LINE 197-198: By the end of follow-up, 133,322 probands (26.0%) were exposed, i.e. had a sibling with an infection-related hospitalization (Fig 1).

Reviewers' comments:

Reviewer's Responses to Questions

Comments to the Author

1. Is the manuscript technically sound, and do the data support the conclusions?

Reviewer #1: Yes

2. Has the statistical analysis been performed appropriately and rigorously? 

Reviewer #1: Yes

3. Have the authors made all data underlying the findings in their manuscript fully available?

Reviewer #1: Yes

4. Is the manuscript presented in an intelligible fashion and written in standard English?

Reviewer #1: Yes

5. Review Comments to the Author

Reviewer #1: This is a well-written manuscript for a well-conducted study. The manuscript may benefit from revisions as indicated in the comment bubbles inserted in the attached files to improve the clarity and impact on intended audience.

1. Introduction line 58: Healthcare costs may not be comparable between US and Australia. Are there similar results reported in Australia? If not, it may be useful to call that out.

RESPONSE: We would like to thank Reviewer 1 for their helpful input. We are not aware of similar estimates in Australia and have added the below underlined section to address this:

PAGE 3, LINES 55-58: US national estimates from 2000-2012 indicate that infections in children aged <19 years accounted for 24.5% of all hospitalizations.[1] In 2003, 43% of US children aged <1 year admitted to hospital had an infection-related diagnosis, incurring a total cost of $690 million.[2] No analogous current data are available from Australia.

2. Introduction line 71: How about sex as a potential confounder?

RESPONSE: We consider sex as potential confounder and controlled for it in the adjusted models. See section from statistical methods below:

Multivariable analyses adjusted for maternal age (<20, 20-24, 25-29, 30-34, ≥35 years), parity (previous births no/yes), gestational age (<28, 28-29, 30-31, 32-34, 35, 36, 37, 38, 39-40, 41, ≥42 weeks), birth weight (1000-1499, 1500-1999, 2000-2499, 2500-2999, 3000-3499, 3500-3999, ≥4000 grams), sex, season of birth (spring, summer, autumn, winter), 5-minute Apgar score (0-7, 8-10), birth mode (vaginal/cesarean), and year (2 year blocks).

In the referenced section (introduction line 71), we discuss the main characteristics of infections among siblings that we explored in this study. Although infection tends to be more common in males, exploring sex differences among siblings was not a priority aim of ours. This is not to say that this should not be studied further in future studies. 

3. Methods line 123: Since infection-related hospitalizations up to first 3 admissions were considered, it is not explained how this factors impacted the results or its association with the outcome.

RESPONSE: The first 3 infection-related admissions were used for both the exposure (i.e. infections in the sibling) and outcome (i.e. infections in the proband). 

For sibling infection as the exposure, the number of sibling admissions was categorized as 0 (unexposed), 1, 2, 3+ and infection risk in the proband was estimated for these different exposure categories. All results present the estimated risk for the different levels of exposure.

For proband infection as the outcome, Cox regression models for recurrent events data were used to account for multiple admissions, up to the first 3 admissions. These proband infection-related hospital admissions (up to the first 3) are the events in the model and analyses are stratified by event order. The final estimate that is presented in the results is the overall effect. 

We have re-written and added the following underlined information to help clarify this in the methods section.

PAGE 5, LINE 114-126: The study outcome was infection-related hospitalization occurring in the child of analysis, termed ‘proband’ hereafter (‘proband infection-related hospitalization’). Up to the first 3 proband infection-related hospitalizations occurring during the follow-up period were considered and analyzed as recurrent events data. 

Exposure was defined as the number of infection-related hospitalizations occurring in a proband’s sibling (‘sibling infection-related hospitalization’), categorized as (0, 1, 2, 3+). The proband became ‘exposed’ once a sibling had an infection-related hospitalization during the follow-up period. Exposure was considered time-varying; once a proband became exposed, they were exposed for the remainder of the follow-up period and the level of exposure at the time of each outcome, was the number of sibling infection-related hospitalizations that occurred prior to that outcome. Probands were ‘unexposed’ for the period preceding the first sibling infection-related hospitalization or if no sibling infection-related hospitalizations occurred during follow-up. 

PAGE 7, LINES 150-155: Adjusted hazard ratios (HRs) and 95% confidence intervals (CIs) were estimated using multivariate Cox proportional hazard regression models for recurrent events data (conditional risk set models) by Prentice et al.[15] Proband infection-related hospitalizations were the events and age in months was the underlying time variable. Time to each event, up to the first 3 admissions, was measured from entry time, and analyses were stratified by event order with the final estimate as the overall effect.

4. Methods line 130: What about children who may be presently living with their fathers only? Or children living in single-parent household.

RESPONSE: We appreciate the reviewer’s comment and also considered this eventuality. Unfortunately, however, we do not have information on whom the children were living with or if they were from single parent households. Such data would be difficult to interpret as the time spent in different locations would be impossible to quantify in a large study. 

We are not aware of any research that suggest household structure may affect the child’s infection risk unless it is based on family size. We have mentioned in the discussion that these data were not available. 

PAGE 19, LINES 350-353: Data on infections managed in primary care or in emergency departments and data on potential unmeasured co-variates, (e.g. child care attendance, breastfeeding, tobacco smoke exposure, household structure, obesity, parental chronic disease, and environmental exposures) were unavailable.

5. Figure 1 is not referenced in the text.

RESPONSE: We have now referenced Figure 1 in the results section.

6. Results line 190: “By the end of follow-up, 133,322 probands (26.0%) were exposed, i.e. had a sibling with an infection-related hospitalization”

Percentage could be presented out of population of interest (142,915) rather than total no. of probands. It appears that ~93% are exposed making the study population heavily skewed towards exposed probands.

RESPONSE: The 133,322 refers to the total number of probands who were exposed by the end of follow-up whereas the 142,915 refers to the total number of probands who had an infection-related hospitalization, i.e. the number who had the outcome. The population of interest, i.e. the population that was followed, is the 512,279 probands and not the 142,915. Therefore the 133,322 is not a proportion of the 142,915 since some of those who were exposed did not have the outcome by the end of follow-up. To clarify these numbers, we have revised Figure 1 and Table 1 to include the breakdown of the population by exposure and outcome. 

Figure 1 – Flowchart of study population

N=710,320 

live-born children 1980-2014 

n=174,203 children excluded because of either:

incomplete date of birth, no recorded full siblings during the study period or siblings included twins or higher multiples, or turned 18 before siblings entered study period

n=536117 

n=23,838 children excluded because of Aboriginal or Torres Strait Islander descent

n=512,279

final study population 

*Time-varying exposure status at end of follow-up period

Table 1 (only showing added information in italics)

 Infection-related hospitalization in sibling(s) (Exposure) Total study population

 No (Unexposed) Yes (Exposed) 

 Characteristic N % N % N %

 378,957 74 133,322 26 512,279 100

Total # of infection-related hospitalizations during study period in proband 

0 263,982 69.7 105,382 79.0 369,364 72.1

1 73,930 19.5 19,403 14.6 93,333 18.2

2 23,469 6.2 5,273 4.0 28,742 5.6

≥3 17,576 4.6 3,264 2.5 20,840 4.1

7. Results line 192: “The median age of when probands were exposed was 4.7 years (interquartile range, IQR 2.7-7.7) among those who did not have an infection-related hospitalization during follow-up, and 4.2 (IQR 2.2-7.1) among those who did.”

How is this relevant for the current analysis?

RESPONSE: These descriptive statistics are intended to show that the median age of exposure was similar among all probands. 

8. Table 1: What is the size of the study population? 142,915 who had at least one infection-related hospitalization or or 512,279 which is the total no. of probands in WA during the study period? Based on the stated objectives, it seems like the study population is 142,915 probands who had at least one infection-related hospitalization.

RESPONSE: We apologize that this is not clear in the manuscript. The study population is 512,279, which is the total number of probands who were followed. Out of these, 142,915 had the outcome by the end of follow-up. Some of the confusion might have been due to the use of the term “subsequent risk” in the manuscript. We used “subsequent” to refer to the risk following the sibling’s infection but can now see that it might have been interpreted as risk for a second infection in the proband. We have therefore removed the word “subsequent” and made the below changes to clarify this important point. 

PAGE 2, LINE 23 (Abstract): We hypothesized that having siblings hospitalized for infection would increase the proband’s subsequent risk of admission with infection.

PAGE 3, LINES 68-69 (Introduction): We hypothesized that having siblings hospitalized for infection would increase the proband’s subsequent risk of admission with infection.

PAGE 9, LINES 194-198: Of the 512,279 probands in the study population, 369,364 (72.1%) did not have an infection-related hospitalization and 142,915 (27.9%) had at least one infection-related hospitalization, of which 49,582 (9.7% of the study population) had multiple infection-related hospitalization. By the end of follow-up, 133,322 of the study population (26.0%) were exposed, i.e. had a sibling with an infection-related hospitalization (Fig 1).

9. Table 1: Among 512,279 probands in the population, only cases (probands with at least one infection-related hospitalization, n = 142,915 can have the exposure i.e. infection-related hospitalization in sibling(s). None of the controls can be exposed?

RESPONSE: Sorry for the confusion. Both cases (probands with at least 1 infection-related hospitalization, n= 142,915) and non-cases (probands without any infection-related hospitalization, n=369,364) could have been exposed. We have changed the header in table 1 from “total” to “total study population” and added the section for “Total # of infection-related hospitalizations during study period in proband” to help clarify.

Table 1 

 Infection-related hospitalization in sibling(s) (Exposure) Total study population

 No (Unexposed) Yes (Exposed) 

 Characteristic N % N % N %

 378,957 74 133,322 26 512,279 100

Total # of infection-related hospitalizations during study period in proband 

0 263,982 69.7 105,382 79.0 369,364 72.1

1 73,930 19.5 19,403 14.6 93,333 18.2

2 23,469 6.2 5,273 4.0 28,742 5.6

≥3 17,576 4.6 3,264 2.5 20,840 4.1

10. Table 1: Why so many categories in gestational age unless there is a strong rationale for association between gestational age among premature babies and risk of severe infections

RESPONSE: In our previous work in a similar population (PMID: 27052469), we found that infection-related hospitalization rates in children increased by 12% for each week reduction in gestational age less than 39-40 weeks and by 19% for each 500g reduction in birthweight less than 3000-3500g. Given these observed associations between gestational age and birthweight with severe infection in children, we chose to use the multiple categories to adequately control for potential confounding.

Miller JE, Hammond GC, Strunk T, Moore HC, Leonard H, Carter KW, et al. Association of gestational age and growth measures at birth with infection-related admissions to hospital throughout childhood: a population-based, data-linkage study from Western Australia. Lancet Infect Dis. 2016. doi: 10.1016/S1473-3099(16)00150-X. PubMed PMID: 27052469.

11. Table 1: Same comment as before (for birthweight)

RESPONSE: As indicated above, we observed increased rates of severe infection for each 500g reduction in birthweight. We therefore chose to include the multiple categories of birthweight in the analyses.

12. Table 1: Why is maternal socioeconomic status considered as a confounder but not family socioeconomic status? Single-parent vs. two parents may also be a confounder?

RESPONSE: We used the term ‘maternal’ since the address at birth, which corresponds to the mother regardless of relationship status, was used to define socioeconomic status from the socioeconomic census data. Throughout the text we use the term ‘socioeconomic status’ or ‘SES’. To be consistent and to reduce confusion, we have changed the label in Table 1 to ‘Socioeconomic status’. 

We are not aware of any research that suggests single parent vs two parents may confound the overall association, but as it is a plausible suggestion, we have added that data on household structure were not available:

PAGE 19, LINES 350-353: Data on infections managed in primary care or in emergency departments and data on potential unmeasured co-variates, (e.g. child care attendance, breastfeeding, tobacco smoke exposure, household structure, obesity, parental chronic disease, and environmental exposures) were unavailable.

13. It is not very clear how maternal socioeconomic status is derived from SEIFA data. 

RESPONSE: We have expanded the below text in the methods section to help clarify how we derived maternal socioeconomic status, which we have renamed ‘socioeconomic status’.

PAGE 6, LINES 130-135: Area-level socioeconomic status (SES) was derived from Socio-Economic Indexes for Areas (SEIFA), which are summary measures of socioeconomic variables associated with disadvantage at the census Collection District level. The indexes can be used to rank collection districts according to the general socioeconomic wellbeing of residents. Percentiles for SES were defined by matching address at birth to the SEIFA score for the same census Collector’s District from the census year closest to the birth year. [14]

Pink B. An Introduction to Socio-Economic Indexes for Areas (SEIFA). In: Statistics. ABo, editor. Canberra: Australian Bureau of Statistics.; 2006.

14. Similar trend is observed in the adjusted hazard ratios across all clinical types of infections. Table 2 may be represented as a figure or part of supplementary material.

RESPONSE: As per the reviewer’s suggestion, we have turned Table 2 into a figure (now Fig 3) below. We believe that the data are of interest and add to previous studies as we have a large enough sample size and a priori coding of infections to present by infection type. This granularity is unusual as most studies use an incomplete categorization of infections as the outcome, or simply report overall infections. We therefore believe that this figure warrants inclusion in the main body of the manuscript, rather than in a supplementary online file.

Fig 3. Infection-related hospitalization sibling hazard ratios by clinical infection groups.

15. Line 229: S1 Fig. shows adjusted risk rations and not adjusted hazard ratios. Improve consistency in text reporting.

RESPONSE: We apologize for this inconsistency and have changed the label to correctly state that these estimates are hazard ratios. 

16. It may be useful to explain how the results from this study impact public health measures in WA.

RESPONSE: We have added the following text to the discussion:

PAGE 21, LINES 378-384: The study, which may be broadly applicable to other high-income settings, highlights the increased risk in siblings of children hospitalized for infection. Interventions for other family members in this context is rarely considered in clinical pediatric practice (beyond specific infections, such as antibiotic prophylaxis for meningococcal infection). These findings suggest that simple interventions, such as promoting breast-feeding of younger siblings and timely and complete vaccination may be particularly pertinent in families where a child has been hospitalized with an infection.

17. Given the data collection ended 6 years ago, it will be useful to include a discussion regarding any potential drift in healthcare practice patterns in the last 6 years that may influence the results of this study.

RESPONSE: This is a good point and partly reflects the delays in accessing population linked data. There have been few changes in healthcare practice since 2014 and the following has been added to the discussion to address this important point:

PAGE 20, LINES 370-377: Changes in healthcare practice in Australia since the end of the study in 2014 are unlikely to have influenced study results. Apart from universal free influenza immunization for children aged 6 months to 5 years in 2018, there have been few changes to public health interventions that would affect the data. Other changes to immunization policy, such as introduction of meningococcal B vaccine are unlikely to have impacted the findings as the incidence is very low. Expansion of hospital in the home antibiotic therapy (outpatient antibiotic therapy in the US) has expanded in the last few years, but these children are still classified as hospital admissions and so the outcome data would not be affected.

18. Figure 1 - It is not clear if all of 512,279 probands had an infection-related hospitalization. As per text on Page 9, line 188, only 142,915 probands had at least one infection-related hospitalization

RESPONSE: We apologize for the confusion and have added exposure status to Figure 1. We have also reworded the referenced text to help with clarity:

PAGE 9, LINES 194-198: Of the 512,279 probands in the study population, 369,364 (72.1%) did not have an infection-related hospitalization and 142,915 (27.9%) had at least one infection-related hospitalization, of which 49,582 (9.7% of the study population) had multiple infection-related hospitalization. By the end of follow-up, 133,322 of the study population (26.0%) were exposed, i.e. had a sibling with an infection-related hospitalization (Fig 1).

19. Because non-cases can never be exposed, the objective of showing non-cases in Figure 3 is unclear and is unnecessarily complicating its interpretation.

RESPONSE: We have tried to clarify the text since both cases and non-cases could be exposed. Figure 3 is therefore relevant because it shows the distribution of cases and non-cases who were exposed based on their time at risk. 

20. Strobe checklist, item #1: It may be useful to indicate in the title that the study assesses familial risk of infection-related hospitalization in children/pediatric population.

RESPONSE: We have changed the title to the below:

The familial risk of infection-related hospitalization in children: a population-based sibling study 

21. Strobe checklist – variables, item #7: It’s unclear if only first infection-related hospitalization is considered for outcome or subsequent hospitalizations are also considered in the proband as indicated in the abstract.

RESPONSE: We realize that the term “subsequent risk” has resulted in some confusion and have removed the word “subsequent”. We have added to Table 1 and the study flowchart to help clarify the confusion.

---

## [Editor Report · Decision Letter 1]

1 Apr 2021

The familial risk of infection-related hospitalization in children: a population-based sibling study

PONE-D-20-33021R1

Dear Dr. Miller,

We’re pleased to inform you that your manuscript has been judged scientifically suitable for publication and will be formally accepted for publication once it meets all outstanding technical requirements.

Kind regards,

Sreeram V. Ramagopalan

Academic Editor

PLOS ONE
---

## [Editor Report · Acceptance letter]

7 Apr 2021

PONE-D-20-33021R1 

The familial risk of infection-related hospitalization in children: a population-based sibling study 

Dear Dr. Miller:

I'm pleased to inform you that your manuscript has been deemed suitable for publication in PLOS ONE. Congratulations! Your manuscript is now with our production department. 

Kind regards, 

on behalf of

Dr. Sreeram V. Ramagopalan 

Academic Editor

PLOS ONE